# Targeted Study of the Effect of Yeast Strain on Volatile Compounds Produced in Sorghum Beer

**DOI:** 10.3390/foods13223626

**Published:** 2024-11-14

**Authors:** Drew Budner, Joseph Carr, Brett Serafini, Samantha Tucker, Elisabeth Dieckman-Meyer, Lindsey Bell, Katherine A. Thompson-Witrick

**Affiliations:** 1Department of Chemistry, Coastal Carolina University, P.O. Box 261954, Conway, SC 29528, USA; 2Department of Mathematics & Statistics, Coastal Carolina University, P.O. Box 261954, Conway, SC 29528, USA; lbell2@coastal.edu; 3Food Science and Human Nutrition Department, University of Florida, Gainesville, FL 32611, USA; kthompsonwitrick@ufl.edu

**Keywords:** sorghum, malt, GCMS, yeast, volatile organic compounds

## Abstract

An increase in the consumer demand and the availability of gluten-free products has led to several brewers investigating brewing with grains other than barley. The primary grain of choice has been sorghum. These new gluten-free beers have a unique flavor and aroma, which previous research has shown is the result of differences in concentration for key chemical compounds, including ethyl butyrate, butyl acetate, isoamyl acetate, ethyl caproate, hexyl acetate, 1-octanol, nonanal, ethyl octanoate, and ethyl decanoate. This study focused looked at the influence different strains of yeast had on the concentration of these key compounds. Beer was brewed using either barley or sorghum malt extract. The concentrations of these key volatile compounds were determined using Solid Phase Microextraction (SPME) with Gas Chromatography and Mass Spectral (GC-MS) detection. Overall, it was found that the concentrations of these compounds were statistically different in the beers brewed from these two grain types. However, the yeast strain had no significant impact on the concentrations.

## 1. Introduction

While beer is an extremely popular beverage, for the small portion of the population who are diagnosed with the autoimmune disease celiacs disease, it is recommended to not consume beer or other gluten containing products due to the damage inflicted upon the small intestine [1]. Due to adherence to a strict gluten free diet, these individuals are often forced to miss out on certain social activities such as grabbing a beer with friends [2]. Beer is traditionally made utilizing malted barley or other grain sources that contain gluten. The safest thing for someone who has a gluten-sensitivity or celiacs disease would be to consume beers that are made using gluten-free grains such as sorghum, millet, rice or any of the pseudograins (buckwheat, amaranth, or quinoa) [3].

Beer is one of the most popular beverages consumed throughout the world [4]. Beer is produced through the fermentation process, resulting in the formation of a number of complex compounds [5]. The formation of flavor compounds in beer plays a significant role in the quality of the final product. Flavor compounds not only influence aroma characteristics, but the taste as well [6,7]. Beer consumers will judge the quality of beer based upon the balance of the volatile and semi-volatile compounds found within that beer [7]. These compounds are produced through a number of chemical reactions throughout the malting, brewing, and fermentation process [8]. Research has focused on the impact of ingredients, such as wort’s composition [8,9,10], grain type [11,12], fermentation conditions, and the yeast strain selected [13]. A brewers’ ability to control the production of specific flavor compounds is incredibly important for the flavor of the beer. The aromatic compounds found in beer generally fall into one of six categories: higher aliphatic and aromatic alcohols, multivalent alcohols, esters, carbonyl compounds, sulfur-containing compounds, and organic acids [14,15]. The volatile compounds found in the beer must be carefully balanced and if that balance is thrown it could negatively affect the flavor and aroma composition of the beer [16,17].

Over 750 million people live in the semi-tropical and tropical regions of Africa (Burkina Faso, Ethiopia, Nigeria, and Sudan), Asia (China and India), and Central and South America, where sorghum plays an important role in their everyday lives, along with corn, rice, wheat, and barley [18,19,20]. In 2023/2024, over 58.28 metric tons of sorghum was produced globally with the United States (8.07 million), Nigeria (6.4 million), India (4.74 million), Mexico (4.49 million), and Brazil (4.43 million) rounding out the top five producers [21]. Similarly to barley and corn, sorghum falls into the *Poaceae* family. Sorghum, like corn, shares similar characteristics in terms of plant shape and genomic organization [22,23]. Unlike some other crops, sorghum is more drought tolerant, which makes it an ideal crop for semi-arid regions of the world. In the United States, sorghum is predominantly used as animal fodder [19]; however, with the growing gluten-free food and beverage industry, sorghum is starting to receive more attention. In 2023, the global Gluten-Free Alcoholic Drink market was estimated to be valued at 5.6 billion USD. Over the next decade it is estimated to grow at CAGR (compound annual growth rate) 8.5%, estimated to be valued at 12.7 billion USD by 2033 [24].

One of the most commonly used gluten-free grains is sorghum. Sorghum beers have been brewed in many countries in Africa, typically using mixed fermentations. These beers are typically more viscous, and the mixed fermentations result in a sour beverage with some sweetness. The use of sorghum typically also produces a beer that is more yellow. There are two major differences between sorghum and barley: (1) Sorghum has a significantly higher gelatinization temperature, with 69 °C as the peak temperature. (2) Sorghum a lower β amylase activity in the malt compared to barley. In addition, there are a large number of sorghum varieties that tend to be more waxy, which results in higher amylopectin and low amylose. This makes sorghum more susceptible to hydrolysis from amylolytic and proteolytic enzymes [2]. Brewers have found ways to overcome these challenges, including mashing at higher temperatures and the addition of thermally stable enzymes. This has allowed sorghum, a popular gluten-free grain, to be used by the brewing industry.

The majority of the work reported in the literature has focused on either explorations of traditional African beverages or the use of sorghum as an adjunct. Therefore, this work, focusing on the use of sorghum as the entire grain bill, is unique and has yet to be fully investigated. Prior research into sorghum beer from our lab focused on comparing the aromatic profile of sorghum beer to malted barley using a single strain yeast with no hop addition. The objective of this study was to focus on analyzing the aromatic profile of sorghum beer compared to malted barley beer made utilizing different strains of brewer’s yeast using SPME-GC-MS.

## 2. Materials and Methods

*Chemicals*—All chemicals were of the highest purity available and used as received without further treatment or purification. Sodium chloride was purchased BDH (Radnar, PA, USA). Ethyl caproate, 1-heptanol, and guaiacol were purchased from TCI (Tokyo, Japan). Regarding the other key compounds: ethyl butyrate, butyl acetate, isoamyl acetate, ethyl caproate, hexyl acetate, 1-octanol, nonanal, ethyl octanoate, and ethyl decanoate, were purchased from Alfa Aesar (Haverhill, MA, USA).

*Yeast Propagation*—For the targeted study, four different yeast strains: Wyeast 1056 (American Ale), Wyeast 1098 (British Ale), Wyeast 1010 (America Wheat), and Wyeast 1214 (Belgian Abbey) were used. These strains were selected to provide a range of possible style implications. Prior to the start of the experiment, yeast was propagated using 300 mL of either sorghum or amber malt wort. The propagation wort was created using 40 g of either Maillard Malts^®^ Amber Extract Syrup or Maillard Malts^®^ Sorghum Extract Syrup in 300 mL of distilled water. The extract was brought to boil to destroy any potential microorganisms that might have come in contact with it. The wort was cooled down and inoculated with one of four yeast strains. All liquid yeast slurries were pitched directly into the wort. Yeast cell counts and viability were not performed prior to pitching.

As previously mentioned, four different yeast strains were selected because they encompass a variety of different beer styles commonly found on the market. Table 1 describes each yeast strain based upon their ability to flocculate, ester production, and examples of the beer styles that use these yeasts.

Yeast flocculation is based upon the yeast’s ability to aggregate together and form a multi-cellular organism that will then fall out of suspension. The manufacturer ranked the yeast’s ability to flocculate based upon a 3-pt scale (low, medium, and high). High flocculating yeast strains are the easiest to flocculate, which unfortunately can leave behind unfermented sugars and potentially some unwanted flavor compounds such as diacetyl. Medium flocculation yeast strains are generally used more often due to their ability to stay in suspension longer. Medium flocculation yeast tends to fall out of suspension when the fermentable sugars become less available. Low flocculation yeast strains, unlike highly flocculant yeast, will stay in suspension well after the fermentation has been completed, resulting in a “hazy” or “less clear” beer. Wheat beers commonly utilize low flocculating yeast [25].

*Brewing*—When brewing each beer, approximately 23 L of deionized (DI) water was brought to a boil using a 10 gal Worthog electric brewing system. To this, 2.72 kg of Maillard Malts^®^ Sorghum extract syrup (for sorghum beers) or Maillard Malts^®^ Amber Malt extract syrup (for barley beers) was added to the boil. The mixture was then boiled for 60 min and then cooled before the volume was adjusted to 19 L with DI water. The wort was then transferred into four separate clean and sterile fermenters. The associated yeast slurry was pitched, and the fermenters sealed with an airlock. Fermentation occurred at room temperature over a two-week period. Samples were collected into a clean headspace vial before yeast pitching, and on days 3, 7, and 14. For each malt type (barley and sorghum) three separate brews were performed.

*Volatile Analysis*—The selected compounds were analyzed using the GC-MS method described in Budner et al. (2021) [10].

*Compound Quantification*—A set of nine compounds which have been found to be present in beers brewed from both malt and malted sorghum at statistically different amounts were selected for quantification (Table 2) [11]. The compounds (ethyl butyrate, butyl acetate, isoamyl acetate, ethyl caproate, hexyl acetate, 1-octanol, nonanal, ethyl octanoate, and ethyl decanoate) were quantified using the calculation of response factors, calculated by evaluation of a set of linear calibration standards of concentrations of 1.00, 100, 200, 300, 400, and 500 mg/L (Table 2).

*Statistics*—Concentrations were analyzed on the log-scale due to a heavy right skew in the data. Repeated measure ANOVA with unstructured correlation was computed using the nlme package in R version 4.3.3. [26]. Using the fitted model, planned contrasts were computed to compare the difference in average concentration between barley and sorghum at each of the four time points [27]. Contrasts were additionally computed to compare the average concentration of each grain on the initial day of brewing with the final day [28].

## 3. Results and Discussion

There is a growing demand for new and improved gluten-free beers. The sensory characteristics of traditionally made malted barley are well established, especially when it comes to volatile organic compounds (VOCs) [29]. Previous investigations between beers brewed with malt extract or sorghum extract indicate that the differences in chemical composition of the volatile and semi-volatile compounds are primarily of variable concentrations, not composition. In addition, a set of key compounds that differ were identified [11]. Reported here was the investigation of the changes in the concentration of these key volatile and semi-volatile compounds produced by different yeast strains when placed in either barley or sorghum wort.

*Compounds Selected:* The analysis of a large number of replicate brews using sorghum or barley malt showed that there were a limited number of unique compounds between the two produced beers. However, there were several compounds that were present in both beers but at significantly different concentrations [11]. A subset of these compounds was selected for investigation here, including ethyl butyrate, butyl acetate, isoamyl acetate, ethyl caproate, hexyl acetate, 1-octanol, nonanal, ethyl octanoate, and ethyl decanoate. The concentration of each of these in beer brewed from either sorghum or barley taken pre-pitch, and at fermentation day 3, 7, 14 are shown in Table 1.

*Impact of grain (barley* vs. *sorghum):* Overall, there was not a statistical difference between the concentration of the key compounds before fermentation occurred. However, if you look at individual compounds, there are apparent differences. At this stage in the brewing process, all the volatile compounds come from the grain and any hop additions. The yeast itself has not been added to the wort, and thus would not contribute to the aromatic profile. The most significant change in the concentration of the volatile profile for the two beers (sorghum and barley) was between day 0 and 3. This is a result of the fermentation process starting and the yeast starting to produce primarily (ethanol) and secondary metabolites (volatile compounds) [16].

As the fermentation progresses, the concentration of the volatile compounds will start to increase [14]. While the Free Amino Nitrogen (FAN) levels were not measured in this experiment, it has been shown that sorghum produces wort with lower FAN. The low availability of nitrogen can impact yeast viability and reduce the fermentation potential of the wort. This appears to be the case in these sorghum beers [11]. The results from previous research are similar to other studies that also showed that the resulting wort contained lower FAN levels than the corresponding malted barley [30].

As shown in Figure 1, statistical differences (*p* < 0.001) were observed between the two different grain types for days 3, 7, and 14. No statistical differences (*p* < 0.2483) were observed between the two grain types on day 0. In each of the subsequent days, barley had a significantly higher concentration than sorghum (*p*-values < 0.0001) (Table 3). However, a similar increase in concentrations was observed in both barley and sorghum fermentations. This is expected, as the yeast produces secondary metabolites [10]. It appears that fermentation is occurring as expected in both beer types. However, based on the smaller increase in volatile and semi-volatile secondary metabolites, it occurs at a significantly lower rate in the sorghum beers compared to the barley. This results in the statistically different concentrations observed. Ma et al. (2016) investigated the impact of both extruded and non-extruded sorghum as an adjunct in beer show significant concentrations of ethyl butyrate, ethyl caproate, ethyl octanoate, and ethyl decanoate [5].

*Impact of yeast:* While fermentation occurs in both sorghum and barley beers, the impact of yeast was investigated. As the different yeast strains have been shown to be critical in producing key characteristics of different beer styles, it is of interest to determine if the impact of these yeast strains is impacted by grain type. As shown in Figure 2, there were no statistically significant differences observed between sorghum and barley beers brewed with these different yeast strains. For the malted barley and sorghum, the concentrations appear to follow a similar trend, with concentrations increasing with fermentation time. It is of interest that for these selected compounds, there is no statistically significant impact. This could be a result of the metabolomic pathway responsible for the production of these specific compounds that are common to all these yeast strains. Einfalt (2021) investigated the impact of various yeast strains on beers utilizing sorghum as a brewing adjunct. It was reported that there were significant variations in the measured volatile compound concentrations; however, none of compounds studied in this work were included in the report [31]. It would be of interest to perform a non-targeted investigation to determine if these yeast strains have any significant impact on the volatile or semi volatile compositions.

## 4. Conclusions

An investigation into the impact of four different yeast strains on the concentration of ethyl butyrate, butyl acetate, isoamyl acetate, ethyl caproate, hexyl acetate, 1-octanol, nonanal, ethyl octanoate, and ethyl decanoate was undertaken. As previously reported, the concentrations of these compounds were found to be statistically different because of fermentation. However, there was no observed impact on the concentration of these compounds due to the yeast strains investigated. Therefore, we would not expect significant differences in these key compounds based on yeast strain. However, it would be of interest to expand the investigation into additional volatile or semi-volatile compounds.

## Figures and Tables

**Figure 1 foods-13-03626-f001:**
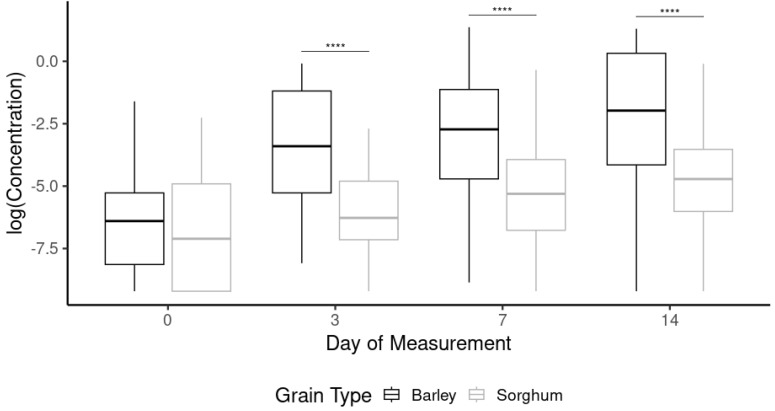
Box plot comparison of the concentration (log) of volatile compounds for all the yeast strains combined to the day of measurement. The concentration was expressed as log (concentration) to provide better visualization of the impact of variables. The **** indicates the difference is highly significant.

**Figure 2 foods-13-03626-f002:**
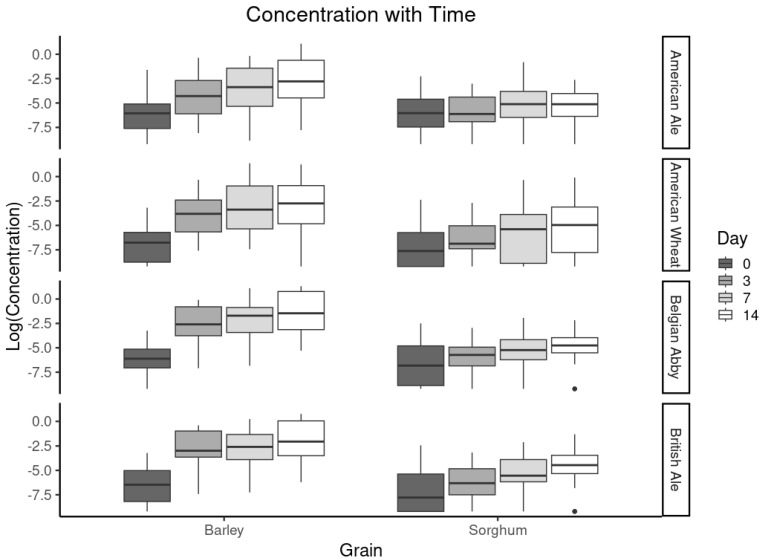
Box plot comparison of concentration (log) of selected volatile compounds over time (days 0, 3, 7, 14). The concentration was expressed as log (concentration) to provide better visualization and impact of variables. The dots are the results of compounds that were not detected as included in the sample set, producing outlier data points.

**Table 1 foods-13-03626-t001:** Yeast Characteristics of the Four Strains Utilized.

Yeast Strain	Flocculation	Ester Production	Beer Styles Used
American Ale	Low/Medium	Medium	American Pale Ale, American Amber Ale
British Ale	Medium	Low	English IPA, British Golden Ale
American Wheat	Low	Low	American Wheat Ale, Kölsch
Belgian Abbey	Low/Medium	Medium	Belgian Tripel, Belgian Dark Strong Ale

Table was created utilizing the specification data from the manufacturer’s website (https://wyeastlab.com (accessed on 12 October 2024)).

**Table 2 foods-13-03626-t002:** The nine compounds of interest with the determined linear retention index (LRI) values compared to the aliphatic hydrocarbon along with the reported flavor descriptors.

Compound	LRI	Flavor Descriptor
ethyl butyrate	778	Pineapple
butyl acetate	771	Ethereal
isoamyl acetate	873	Sweet, banana
ethyl caproate	1000	Fruity, sweet
hexyl acetate	990	Apple peel
1-octanol	1054	Aromatic/chemical
nonanal	1105	Fat, citrus, green
ethyl octanoate	1348	Fruity
ethyl decanoate	1360	Grape

**Table 3 foods-13-03626-t003:** The measured concentration of the identified key aroma compounds. The concentrations were determined at four different times, day 0 (pre-pitch), day 3, day 7, and day 14.

**Day 0**	**American Ale**		**Belgian Abby**		**British Ale**		**American Wheat**	
**Compound**	**Barley**	**Sorghum**	**Barley**	**Sorghum**	**Barley**	**Sorghum**	**Barley**	**Sorghum**
ethyl butyrate	2.94 (±5.1) × 10^−3^	4.25 (±2.8) × 10^−3^	5.40 (±3.2) × 10^−3^	4.59 (±2.4) × 10^−3^	3.21 (±3.4) × 10^−3^	5.84 (±2.6) × 10^−3^	4.48 (±1.0) × 10^−3^	2.15 (±2.7) × 10^−3^
butyl acetate	9.92 (±9.1) × 10^−2^	4.95 (±4.9) × 10^−2^	1.86 (±1.3) × 10^−2^	3.90 (±3.7) × 10^−2^	1.86 (±6.4) × 10^−2^	4.02 (±4.1) × 10^−2^	1.92 (±0.8) × 10^−3^	1.67 (±4.4) × 10^−2^
isoamyl acetate	9.22 (±6.5) × 10^−4^	2.53 (±2.8) × 10^−3^	6.33 (±1.0) × 10^−3^	3.16 (±2.7) × 10^−4^	6.33 (±6.5) × 10^−3^	7.99 (±1.3) × 10^−4^	6.72 (±1.1) × 10^−5^	9.62 (±9.8) × 10^−5^
ethyl caproate	8.20 (±4.3) × 10^−4^	5.61 (±3.7) × 10^−4^	5.54 (±4.9) × 10^−4^	3.12 (±4.1) × 10^−4^	5.54 (±1.9) × 10^−4^	3.07 (±3.2) × 10^−4^	2.84 (±3.6) × 10^−4^	8.17 (±4.7) × 10^−5^
hexyl acetate	2.09 (±2.4) × 10^−3^	n.d.	1.44 (±2.3) × 10^−3^	3.25 (±5.6) × 10^−5^	1.44 (±3.9) × 10^−3^	3.11 (±5.3) × 10^−5^	1.03 (±1.1) × 10^−3^	1.94 (±2.2) × 10^−4^
1-Octanol	1.29 (±2.2) × 10^−3^	7.54 (±8.2) × 10^−4^	1.09 (±0.9) × 10^−4^	3.30 (±5.7) × 10^−4^	1.09 (±4.7) × 10^−3^	1.05 (±1.8) × 10^−4^	1.42 (±2.2) × 10^−2^	2.43 (±2.8) × 10^−4^
nonanal	6.45 (±8.5) × 10^−3^	3.45 (±1.7) × 10^−3^	3.26 (±4.6) × 10^−3^	9.03 (±7.8) × 10^−3^	3.26 (±2.6) × 10^−3^	7.60 (±6.8) × 10^−3^	4.19 (±4.3) × 10^−3^	4.70 (±5.7) × 10^−3^
ethyl octanoate	4.06 (±2.9) × 10^−3^	1.35 (±1.3) × 10^−2^	4.21 (±3.5) × 10^−3^	6.33 (±5.5) × 10^−3^	4.21 (±6.6) × 10^−3^	6.15 (±1.0) × 10^−4^	3.20 (±3.8) × 10^−3^	1.15 (±1.1) × 10^−3^
ethyl decanoate	7.46 (±7.1) × 10^−3^	1.66 (±1.5) × 10^−2^	8.89 (±1.1) × 10^−3^	6.49 (±5.8) × 10^−3^	8.89 (±4.4) × 10^−3^	7.41 (±1.2) × 10^−3^	6.51 (±1.0) × 10^−3^	n.d.
Day 3	American Ale		Belgian Abby		British Ale		American Wheat	
Compound	Barley	Sorghum	Barley	Sorghum	Barley	Sorghum	Barley	Sorghum
ethyl butyrate	1.51 (±1.4) × 10^−2^	2.03 (±2.4) × 10^−3^	3.44 (±1.9) × 10^−2^	1.16 (±0.3) × 10^−3^	3.38 (±1.2) × 10^−2^	3.11 (±3.6) × 10^−3^	1.83 (±1.6) × 10^−2^	4.59 (±4.1) × 10^−3^
butyl acetate	3.46 (±2.4) × 10^−2^	3.60 (±2.0) × 10^−2^	6.41 (±6.8) × 10^−2^	2.38 (±2.4) × 10^−2^	3.52 (±1.3) × 10^−2^	2.09 (±1.8) × 10^−2^	5.71 (±5.3) × 10^−2^	7.65 (±1.7) × 10^−3^
isoamyl acetate	2.06 (±2.0) × 10^−1^	6.20 (±5.4) × 10^−3^	6.50 (±2.5) × 10^−1^	4.57 (±1.4) × 10^−3^	5.81 (±0.8) × 10^−1^	1.88 (±0.7) × 10^−3^	4.32 (±3.7) × 10^−1^	3.57 (±1.3) × 10^−3^
ethyl caproate	3.84 (±3.8) × 10^−2^	1.98 (±0.1) × 10^−3^	1.91 (±0.4) × 10^−2^	2.67 (±1.5) × 10^−3^	6.57 (±1.2) × 10^−2^	8.71 (±7.2) × 10^−4^	4.19 (±3.5) × 10^−2^	8.93 (±4.2) × 10^−4^
hexyl acetate	2.71 (±2.3) × 10^−3^	3.37 (±5.8) × 10^−5^	3.91 (±2.0) × 10^−3^	6.43 (±1.1) × 10^−5^	3.91 (±0.5) × 10^−3^	n.d.	3.25 (±2.4) × 10^−3^	n.d.
1-Octanol	1.72 (±1.6) × 10^−2^	1.24 (±1.0) × 10^−3^	4.71 (1.4) × 10^−2^	1.81 (±2.7) × 10^−3^	6.37 (±4.3) × 10^−2^	6.09 (±1.0) × 10^−4^	2.39 (±2.2) × 10^−2^	3.74 (±4.3) × 10^−4^
nonanal	2.83 (±2.9) × 10^−3^	1.50 (±1.2) × 10^−3^	2.64 (±4.1) × 10^−2^	2.73 (±3.4) × 10^−3^	1.07 (±1.3) × 10^−2^	2.60 (±1.4) × 10^−3^	1.53 (±0.8) × 10^−3^	3.38 (±2.4) × 10^−3^
ethyl octanoate	3.24 (±3.5) × 10^−1^	1.89 (±1.5) × 10^−2^	6.61 (±1.3) × 10^−1^	2.20 (±1.5) × 10^−2^	4.03 (±0.2) × 10^−1^	1.62 (±1.5) × 10^−2^	3.34 (±3.1) × 10^−1^	4.80 (±3.3) × 10^−2^
ethyl decanoate	3.27 (±3.4) × 10^−1^	2.15 (±1.8) × 10^−2^	4.42 (±0.7) × 10^−1^	2.21 (±1.9) × 10^−2^	4.06 (±0.6) × 10^−1^	1.15 (±1.0) × 10^−2^	2.96 (±2.5) × 10^−1^	3.31 (±2.2) × 10^−2^
Day 7	American Ale		Belgian Abby		British Ale		American Wheat	
Compound	Barley	Sorghum	Barley	Sorghum	Barley	Sorghum	Barley	Sorghum
ethyl butyrate	1.28 (±0.46) × 10^−1^	8.87 (±8.2) × 10^−3^	1.40 (±0.6) × 10^−1^	6.37 (±6.6) × 10^−3^	6.50 (±6.0) × 10^−2^	2.52 (±0.9) × 10^−3^	7.97 (±9.5) × 10^−2^	9.89 (±3.9) × 10^−3^
butyl acetate	5.76 (±1.0) × 10^−2^	5.12 (±1.7) × 10^−2^	4.38 (±2.1) × 10^−2^	3.28 (±3.6) × 10^−2^	7.91 (±1.7) × 10^−2^	4.37 (±1.1) × 10^−2^	5.08 (±3.9) × 10^−2^	1.60 (±1.2) × 10^−2^
isoamyl acetate	5.06 (±3.82) × 10^−1^	1.35 (±3.6) × 10^−2^	1.27 (±1.5)	9.21 (±1.1) × 10^−3^	7.84 (±5.6) × 10^−1^	1.45 (±4.1) × 10^−2^	1.05 (±1.1)	3.39 (±1.7) × 10^−2^
ethyl caproate	1.74 (±1.6) × 10^−1^	8.55 (±9.2) × 10^−3^	7.34 (±4.5) × 10^−1^	5.68 (±7.0) × 10^−3^	1.12 (±2.0) × 10^−1^	4.44 (±1.7) × 10^−3^	1.84 (±2.0) × 10^−1^	2.06 (±1.2) × 10^−2^
hexyl acetate	3.02 (±4.9) × 10^−3^	1.79 (±2.5) × 10^−4^	1.14 (±1.5) × 10^−1^	1.74 (±3.8) × 10^−4^	3.70 (±4.0) × 10^−3^	2.18 (±7.8) × 10^−4^	1.91 (±1.5) × 10^−2^	4.44 (±4.8) × 10^−4^
1-Octanol	4.22 (±4.2) × 10^−2^	5.15 (±(3.1) × 10^−3^	1.28 (±1.6) × 10^−1^	3.58 (±2.6) × 10^−3^	4.75 (±5.0) × 10^−2^	1.44 (±0.1) × 10^−3^	1.38 (±2.1) × 10^−1^	1.93 (±2.2) × 10^−3^
nonanal	2.47 (±1.3) × 10^−2^	3.55 (±3.3) × 10^−3^	1.27 (±1.9) × 10^−2^	3.95 (±6.6) × 10^−3^	4.65 (±1.4) × 10^−1^	3.28 (±1.0) × 10^−3^	1.64 (±1.8) × 10^−3^	2.51 (±2.0) × 10^−3^
ethyl octanoate	3.60 (±3.6) × 10^−1^	1.56 (±2.5) × 10^−1^	9.45 (±1.3) × 10^−1^	5.87 (±7.4) × 10^−2^	3.86 (±3.4) × 10^−1^	7.27 (±6.1) × 10^−2^	1.52 (±2.1)	3.80 (±3.9) × 10^−1^
ethyl decanoate	2.25 (±2.0) × 10^−1^	6.32 (±9.5) × 10^−2^	7.65 (±7.2) × 10^−1^	3.15 (±4.1) × 10^−2^	3.48 (±1.0) × 10^−1^	5.39 (±4.4) × 10^−2^	7.87 (±1.0) × 10^−1^	1.80 (±1.9) × 10^−1^
Day 14	American Ale		Belgian Abby		British Ale		American Wheat	
Compound	Barley	Sorghum	Barley	Sorghum	Barley	Sorghum	Barley	Sorghum
ethyl butyrate	9.54 (±0.5) × 10^−2^	4.09 (±2.4) × 10^−3^	2.19 (±1.0) × 10^−1^	6.65 (±4.2) × 10^−3^	8.57 (±5.2) × 10^−2^	7.06 (±3.4) × 10^−3^	1.15 (±1.0) × 10^−1^	1.28 (±1.3) × 10^−1^
butyl acetate	5.99 (±9.0) × 10^−2^	4.18 (±2.8) × 10^−2^	4.54 (±0.3) × 10^−2^	3.60 (±1.6) × 10^−2^	6.61 (±3.3) × 10^−2^	2.97 (±1.4) × 10^−2^	5.48 (±3.1) × 10^−2^	3.42 (±3.6) × 10^−1^
isoamyl acetate	1.21 (0.0006)	1.59 (±1.6) × 10^−2^	2.37 (±1.1)	1.15 (±0.7) × 10^−2^	1.37 (±0.8)	4.34 (±3.1) × 10^−2^	1.49 (±1.3)	4.77 (±5.0) × 10^−1^
ethyl caproate	3.62 (±0.0043) × 10^−1^	6.27 (±5.1) × 10^−3^	1.06 (±0.3)	7.06 (±4.0) × 10^−3^	2.50 (±1.7) × 10^−1^	1.30 (±0.8) × 10^−2^	2.66 (±2.3) × 10^−1^	4.60 (±5.0) × 10^−1^
hexyl acetate	7.09 (±2.5) × 10^−3^	1.73 (±2.1) × 10^−4^	1.43 (±0.4) × 10^−2^	3.80 (±6.5) × 10^−4^	1.31 (±0.7) × 10^−2^	3.33 (±5.7) × 10^−4^	1.08 (±0.9) × 10^−2^	1.67 (±1.9) × 10^−4^
1-Octanol	1.43 (±0.022) × 10^−1^	2.32 (±1.6) × 10^−3^	2.38 (±0.3) × 10^−1^	2.67 (±2.4) × 10^−3^	1.78 (±1.2) × 10^−1^	1.74 (±3.1) × 10^−3^	1.28 (±1.1) × 10^−1^	n.d.
nonanal	3.89 (±8.4) × 10^−3^	5.74 (±5.3) × 10^−3^	4.90 (±0.03) × 10^−3^	6.63 (±2.2) × 10^−3^	9.83 (±1.2) × 10^−3^	7.61 (±4.4) × 10^−3^	4.30 (±3.5) × 10^−3^	2.33 (±2.0) × 10^−3^
ethyl octanoate	1.69 (±0.0029)	4.02 (±2.7) × 10^−2^	3.35 (±0.1)	7.43 (±5.3) × 10^−2^	1.57 (±0.6)	1.36 (±1.2) × 10^−1^	2.17 (±1.8)	7.91 (±5.7) × 10^−2^
ethyl decanoate	1.11 (±0.0071)	2.24 (±1.9) × 10^−2^	2.87 (±0.1)	4.14 (±4.3) × 10^−2^	1.62 (±0.4)	7.41 (±5.7) × 10^−2^	1.57 (±1.3)	2.81 (±1.7) × 10^−2^

*N* = 3; mean (±standard deviation) based upon the response value in relation to the internal standard (2-heptanol). n.d.: not detected.

## Data Availability

The original contributions presented in the study are included in the article, further inquiries can be directed to the corresponding author.

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
