# Peer review of "Targeted Study of the Effect of Yeast Strain on Volatile Compounds Produced in Sorghum Beer"

_foods, 2024, doi:10.3390/foods13223626_

Round 1
Reviewer 1 Report
Comments and Suggestions for Authors
This study focused looked at the influence different strains of yeast had on the concentration of these key compounds. The concentrations of these key volatile compounds were determined using Solid Phase Microextraction with Gas Chromatography and Mass Spectral detection. There are several issues in this paper.
(1) The experiment was of simple design.
(2) The compounds identified by mass spectra were confirmed based upon their RI values calculated using nonpolar. All identified components need to be represented in table. CAS, RI, Retention time index, matching degree. Retention indices (RI) must be presented using n-alkane mixtures (C8-C20).
(3) Only nine compounds were discussed. too simple. chemometric methods should be adopted. PLS-DA/PCA/VIP should be added.
(4) Without the future trends.
(5) Please clarify the novelty and contribution of this study more explicitly in Introduction.
(6) What is the sample size? Add the standard deviation of all data.
Comments on the Quality of English LanguageThe English could be improved to more clearly express the research.
Author Response
This study focused looked at the influence different strains of yeast had on the concentration of these key compounds. The concentrations of these key volatile compounds were determined using Solid Phase Microextraction with Gas Chromatography and Mass Spectral detection. There are several issues in this paper.
Comment 1: The experiment was of simple design.
Response 1: We appreciate the comment, we intentionally keep the experimental design simple and straightforward, to make the interpretations meaningful and because the work is primarily done by undergraduate students.
Comment 2: The compounds identified by mass spectra were confirmed based upon their RI values calculated using nonpolar. All identified components need to be represented in table. CAS, RI, Retention time index, matching degree. Retention indices (RI) must be presented using n-alkane mixtures (C8-C20).
Response 2: Thank you for the comment, we have added a table (Table 2) which presents the RI values as well as the flavor descriptors.
Comment 3: Only nine compounds were discussed. too simple. chemometric methods should be adopted. PLS-DA/PCA/VIP should be added.
Response 3: We appreciate the comment and agree that the data and results are straightforward, but we do not feel that the variables measured, and replicates used provide sufficient complexity to provide meaningful results from more complex statistical analysis.
Comment 4: Without the future trends.
Response 4: Thank you for pointing that out. A couple of sentences (lines 64 -67) have since been added to discuss the future growth of the gluten free market including the gluten-free alcoholic drink market.
Comment 5: Please clarify the novelty and contribution of this study more explicitly in Introduction.
Response 5: We have added “The majority of work reported in the literature has focused on either explorations of the traditional African beverages or the use of sorghum as an adjunct. Therefore, this work focuses on the use of sorghum as the entire grain bill is unique and has yet to be fully investigated” to the final paragraph of the introduction.
Comment 6: What is the sample size? Add the standard deviation of all data.
Response 6: We have added a statement in the methods regarding the brewing in triplicated. And have added standard deviations to the table.
Reviewer 2 Report
Comments and Suggestions for Authors
Targeted study of the effect of yeast strain on volatile compounds produced in sorghum beer
Drew Budner, Joseph Carr, Brett Serafini, Samantha Tucker, Elisabeth Dieckman-Meyer, Lindsey Bell, Katherine A. Thompson-Witrick
The study focused on the influence of four different yeast strains on the concentration of nine key volatile compounds in beers made from sorghum and barley. I have some suggestions and questions to the authors before the manuscript is approved for publication.
1. Why ethyl caproate is not included among the investigated key chemical compounds in the Abstract, Compounds Selected and Conclusions, but exists in the tables?
2. Give abbreviation for “Gas Chromatography and Mass Spectral” in the Abstract.
3. The sentences:
“While beer is an extremely popular beverage, there is a small portion of the population who suffers from the autoimmune disease known as celiacs.”
“For the targeted study, four different yeast strains: Wyeast 1056 (American Ale), Wyeast 1098 (British Ale), Wyeast 1010 (America Wheat), and Wyeast 1214 (Belgian Abbey).”
“Previous investigations into the chemical differences between malt and sorghum extract.” should be rewritten and clarified.
4. Please insert in Introduction what the alternatives of sorghum are and why you do not prefer them for this study, also advantages and disadvantages of sorghum need to be elucidated.
5. Clearly indicate the two major differences between sorghum and barley whether facilitate or hinder application in the beer industry.
6. Give examples of the use of the four yeast strains selected: Wyeast 1056 72(American Ale), Wyeast 1098 (British Ale), Wyeast 1010 (America Wheat), and Wyeast 731214 (Belgian Abbey).
7. How do you choose which extract Maillard Malts® Amber Extract Syrup or Maillard Malts® Sorghum Extract Syrup will use? Specify when the first one is used and when the second one is used.
8. Please, include a description of all abbreviations used in the text. The abbreviation on line 106 can be deleted.
9. Number the formula and give explanation for tRN.
10. nmle package in R version 4.3.3 I think should be nlme …. Give a reference for this package.
11. Insert reference on line 178 instead of “(same ref as above)”.
12. Еxplain why the concentration of each component separately is not studied, but the total concentration is used.
13. Think how to present the four tables as figures. Insert figures for clarity.
14. Explain in detail how the nine flavoring ingredients explored in the article were selected.
15. The most references are more than 10 years old. Please insert newly published papers.
Comments on the Quality of English LanguageThe quality of English should be improved.
Author Response
The study focused on the influence of four different yeast strains on the concentration of nine key volatile compounds in beers made from sorghum and barley. I have some suggestions and questions to the authors before the manuscript is approved for publication.
Comment 1: Why ethyl caproate is not included among the investigated key chemical compounds in the Abstract, Compounds Selected and Conclusions, but exists in the tables?
Response 1: Thank you for pointing this out, ethyl caproate was inadvertently left off the list. It has been included.
Comment 2: Give abbreviation for “Gas Chromatography and Mass Spectral” in the Abstract.
Response 2: Thank you for this suggestion the appropriate abbreviation is now included in the abstract.
Comment 3: The sentences:
“While beer is an extremely popular beverage, there is a small portion of the population who suffers from the autoimmune disease known as celiacs.”
“For the targeted study, four different yeast strains: Wyeast 1056 (American Ale), Wyeast 1098 (British Ale), Wyeast 1010 (America Wheat), and Wyeast 1214 (Belgian Abbey).”
“Previous investigations into the chemical differences between malt and sorghum extract.” should be rewritten and clarified.
Response 3: Thank you for this comment, these three sentences have been rewritten to improve clarity.
Comment 4: Please insert in Introduction what the alternatives of sorghum are and why you do not prefer them for this study, also advantages and disadvantages of sorghum need to be elucidated.
Response 4: An additional paragraph (Lines 54 – 67) discuss the importance of sorghum to certain regions of the world and its importance to the gluten-free beverage industry.
Comment 5: Clearly indicate the two major differences between sorghum and barley whether facilitate or hinder application in the beer industry.
Response 5: We have discussed the difference in sorghum and barley in the introduction with the two major differences being the gelatinization temperature and the presence of b amylase. And have addressed how brewers have overcome these.
Comment 6: Give examples of the use of the four yeast strains selected: Wyeast 1056 72(American Ale), Wyeast 1098 (British Ale), Wyeast 1010 (America Wheat), and Wyeast 731214 (Belgian Abbey).
Response 6: Thank you for the comment, we have added a table (Table 1) which includes more information about the yeast selected including examples of uses.
Comment 7: How do you choose which extract Maillard Malts® Amber Extract Syrup or Maillard Malts® Sorghum Extract Syrup will use? Specify when the first one is used and when the second one is used.
Response 7: We are not sure if you understand, only one Extract was used at a time. We did individual brews with only one of these extracts.
Comment 8: Please, include a description of all abbreviations used in the text. The abbreviation on line 106 can be deleted.
Response 8: Thank you for the comments, we have removed the excess definitions of all abbreviations.
Comment 9: Number the formula and give explanation for tRN.
Response 9: Thank you for highlighting the oversite of the equation numbers. However, this section was already explained in reference paper, so it has been removed from the manuscript.
Comment 10: nmle package in R version 4.3.3 I think should be nlme …. Give a reference for this package.
Response 10: Thank you for the comments. We have corrected the name and included appropriate references.
Comment 11: Insert reference on line 178 instead of “(same ref as above)”.
Response 11: Thank you for the comment the correct reference is now included.
Comment 12: Еxplain why the concentration of each component separately is not studied, but the total concentration is used.
Response 12: Unfortunately, we are unclear of what this comment is reference, we only reported individual concentrations and never used total concentration.
Comment 13: Think how to present the four tables as figures. Insert figures for clarity.
Response 13: Thank you for the comment, however, we do not feel that the addition of figures would increase clarity of the data presented.
Comment 14: Explain in detail how the nine flavoring ingredients explored in the article were selected.
Response 14: We feel that the explanation of the method used for the selection of these compounds is adequately described and further information can be found in the included reference. Currently, the manuscript includes this description “
Comment 15: The most references are more than 10 years old. Please insert newly published papers.
Response 15: Thank you for the comment, we have updated the introduction and the discussion and have included several additional newer references.
Reviewer 3 Report
Comments and Suggestions for Authors
The manuscript presents a follow-up work of the same group/laboratory published previously (https://doi.org/10.3390/ beverages7030056) and focused on the comparison between the aromatic profile of sorghum beer and the one made of malted barley using a single strain yeast. The reported results are now completed with data regarding the influence of different yeast strains on beers made of similar raw material (sorghum and malted barley) on the same aromatic profile, using the same analytic approach (SPME-GC-MS). In this regard, the level of novelty and is medium to low, as well as the depth of the research itself.
· Lines 80-82: the information is redundant to Lines 72-74” Wyeast 1056 (American Ale), Wyeast 1098 (Brit- 80 ish Ale), Wyeast 1010 (America Wheat), and Wyeast 1214 (Belgian Abbey)”. Please delete.
· Figure 1 does not provide clear information on each employed commercial yeast starter
· Discussions on similar or close experiments are not sufficiently approached.
· Information provided in the last paragraph of the results is repetitive under Conclusions section
Author Response
The manuscript presents a follow-up work of the same group/laboratory published previously (https://doi.org/10.3390/ beverages7030056) and focused on the comparison between the aromatic profile of sorghum beer and the one made of malted barley using a single strain yeast. The reported results are now completed with data regarding the influence of different yeast strains on beers made of similar raw material (sorghum and malted barley) on the same aromatic profile, using the same analytic approach (SPME-GC-MS). In this regard, the level of novelty and is medium to low, as well as the depth of the research itself.
Comment 1: Lines 80-82: the information is redundant to Lines 72-74” Wyeast 1056 (American Ale), Wyeast 1098 (Brit- 80 ish Ale), Wyeast 1010 (America Wheat), and Wyeast 1214 (Belgian Abbey)”. Please delete.
Response 1: We thank the reviewer for the comment, we agree about the redundancy and have removed this text.
Comment 2: Figure 1 does not provide clear information on each employed commercial yeast starter
Response 2: Thank you for the comment, for figure one we were looking at the impact of fermentation regardless of the impact of the yeast strain. This was specifically to show that the differences between the beers brewed with sorghum were different than the beers brewed with barley, regardless of the impact of yeast strain.
Comment 3: Discussions on similar or close experiments are not sufficiently approached.
Response 3: We have added what limited work we could fine to our discussion, however, we could primarily find reference of the use of sorghum as an adjunct and little to no studies where sorghum is the only component in the grain bill.
Comment 4: Information provided in the last paragraph of the results is repetitive under Conclusions section
Response 4: Thank you for pointing out the repetition, we have cleaned up the conclusion section to reduce this repetitiveness.
Round 2
Reviewer 1 Report
Comments and Suggestions for Authors
accept
Author Response
Comment 1: accept
Response 1: We appreciate the acknowledgement of the improvements made.
Reviewer 2 Report
Comments and Suggestions for Authors
Targeted study of the effect of yeast strain on volatile compounds produced in sorghum beer
The authors have answered more questions, but still exist some unclear points.
1. The sentence: “Previous investigations into the chemical differences between malt and sorghum extract.” is not corrected yet.
2. The abbreviations, such as “DI”, “FAN”, etc. are not described in the text. Please, check carefully for other not elucidated abbreviations and explain them.
3. I understand that only one extract was used at a time - Maillard Malts® Amber Extract Syrup or Maillard Malts® Sorghum Extract Syrup. In addition, you answer that: “We did individual brews with only one of these extracts”, hence you have to clearly explained which one did you use, or maybe you use Maillard Malts® Amber Extract Syrup for barley and Maillard Malts® Sorghum Extract Syrup for sorghum? Please explain in more details this unclarity.
4. If you reported individual concentrations, please indicate in the text that “total concentration” concerns the total concentration of each compound, because “total concentration” can be assumed as total concentration of all compounds.
Comments on the Quality of English LanguageThe English could be improved to more clearly express the research.
Author Response
The authors have answered more questions, but still exist some unclear points.
Comment 1: The sentence: “Previous investigations into the chemical differences between malt and sorghum extract.” is not corrected yet.
Response 1: Thank you for pointing out this oversight. We have removed this fragment.
Comment 2: The abbreviations, such as “DI”, “FAN”, etc. are not described in the text. Please, check carefully for other not elucidated abbreviations and explain them.
Response 2: Thank you for pointing out these undefined abbreviations, we have carefully examined the text and believe we have elucidated all abbreviations.
Comment 3: I understand that only one extract was used at a time - Maillard Malts® Amber Extract Syrup or Maillard Malts® Sorghum Extract Syrup. In addition, you answer that: “We did individual brews with only one of these extracts”, hence you have to clearly explained which one did you use, or maybe you use Maillard Malts® Amber Extract Syrup for barley and Maillard Malts® Sorghum Extract Syrup for sorghum? Please explain in more details this unclarity.
Response 3: We have clarified which extract was used for each beer type in the text.
Comment 4: If you reported individual concentrations, please indicate in the text that “total concentration” concerns the total concentration of each compound, because “total concentration” can be assumed as total concentration of all compounds.
Response 4: We apologize for the misunderstanding of the previous comments. Since we are interested in the apparent differences between the two types of beers, statistically looking at the individual compounds was combined in the ANOVA analysis. The results of this were used to produce the figures. The word total is not a correct representation of what is displayed so we have removed it. Thank you for helping us clarify this aspect
Reviewer 3 Report
Comments and Suggestions for Authors
There are no further comments on the improved manuscript.
Author Response
Comment 1: There are no further comments on the improved manuscript.
Response 1: We appreciate the acknowledgement of the improvements made.